# Practical Hash Functions for Similarity Estimation and Dimensionality Reduction

**Søren Dahlgaard**
University of Copenhagen / SupWiz
s.dahlgaard@supwiz.com

**Mathias Bæk Tejs Knudsen**
University of Copenhagen / SupWiz
m.knudsen@supwiz.com

**Mikkel Thorup**
University of Copenhagen
mthorup@di.ku.dk

## Abstract

Hashing is a basic tool for dimensionality reduction employed in several aspects of machine learning. However, the perfomance analysis is often carried out under the *abstract* assumption that a truly random unit cost hash function is used, without concern for which *concrete* hash function is employed. The concrete hash function may work fine on sufficiently random input. The question is if they can be *trusted* in the real world where they may be faced with more structured input.

In this paper we focus on two prominent applications of hashing, namely similarity estimation with the one permutation hashing (OPH) scheme of Li et al. [NIPS'12] and feature hashing (FH) of Weinberger et al. [ICML'09], both of which have found numerous applications, i.e. in approximate near-neighbour search with LSH and large-scale classification with SVM.

We consider the recent mixed tabulation hash function of Dahlgaard et al. [FOCS'15] which was proved theoretically to perform like a truly random hash function in many applications, including the above OPH. Here we first show improved concentration bounds for FH with truly random hashing and then argue that mixed tabulation performs similar when the input vectors are not too dense. Our main contribution, however, is an experimental comparison of different hashing schemes when used inside FH, OPH, and LSH.

We find that mixed tabulation hashing is almost as fast as the classic multiply-mod-prime scheme $(ax + b) \bmod p$. Mutiply-mod-prime is guaranteed to work well on sufficiently random data, but here we demonstrate that in the above applications, it can lead to bias and poor concentration on both real-world and synthetic data. We also compare with the very popular MurmurHash3, which has no proven guarantees. Mixed tabulation and MurmurHash3 both perform similar to truly random hashing in our experiments. However, mixed tabulation was 40% faster than MurmurHash3, and it has the proven guarantee of good performance (like fully random) on all possible input making it more reliable.

## 1 Introduction

Hashing is a standard technique for dimensionality reduction and is employed as an underlying tool in several aspects of machine learning including search [22, 31, 32, 3], classification [24, 22], duplicate detection [25], computer vision and information retrieval [30]. The need for dimensionality reduction techniques such as hashing is becoming further important due to the huge growth in data sizes. As an example, already in 2010, Tong [36] discussed data sets with $10^{11}$ data points and $10^9$ features.

Furthermore, when working with text, data points are often stored as $w$-shingles (i.e. $w$ contiguous words or bytes) with $w \geq 5$. This further increases the dimension from, say, $10^5$ *common* english words to $10^{5w}$.

Two particularly prominent applications are set similarity estimation as initialized by the MinHash algorithm of Broder, et al. [8, 9] and feature hashing (FH) of Weinberger, et al. [37]. Both applications have in common that they are used as an underlying ingredient in many other applications. While both MinHash and FH can be seen as hash functions mapping an entire set or vector, they are perhaps better described as algorithms implemented using what we will call *basic hash functions*. A basic hash function $h$ maps a given key to a hash value, and any such basic hash function, $h$, can be used to implement Minhash, which maps a set of keys, $A$, to the smallest hash value $\min_{a \in A} h(a)$. A similar case can be made for other locality-sensitive hash functions such as SimHash [12], One Permutation Hashing (OPH) [22, 31, 32], and cross-polytope hashing [2, 33, 20], which are all implemented using basic hash functions.

## 1.1 Importance of understanding basic hash functions

In this paper we analyze the basic hash functions needed for the applications of similarity estimation and FH. This is important for two reasons: 1) As mentioned in [22], dimensionality reduction is often a time bottle-neck and using a fast basic hash function to implement it may improve running times significantly, and 2) the theoretical guarantees of hashing schemes such as Minhash and FH rely crucially on the basic hash functions used to implement it, and this is further propagated into applications of these schemes such as approximate similarity search with the seminal LSH framework of Indyk and Motwani [19].

To fully appreciate this, consider LSH for approximate similarity search implemented with MinHash. We know from [19] that this structure obtains *provably* sub-linear query time and *provably* sub-quadratic space, where the exponent depends on the probability of hash collisions for "similar" and "not-similar" sets. However, we also know that implementing MinHash with a poorly chosen hash function leads to *constant bias* in the estimation [28], and this constant then appears in the *exponent* of both the space and the query time of the search structure leading to worse theoretical guarantees.

Choosing the right basic hash function is an often overlooked aspect, and many authors simply state that any (universal) hash function "is usually sufficient in practice" (see e.g. [22, page 3]). While this is indeed the case most of the time (and provably if the input has enough entropy [26]), many applications rely on taking advantage of highly structured data to perform well (such as classification or similarity search). In these cases a poorly chosen hash function may lead to very systematic inconsistensies. Perhaps the most famous example of this is hashing with linear probing which was deemed very fast but unrealiable in practice until it was fully understood which hash functions to employ (see [35] for discussion and experiments). Other papers (see e.g. [31, 32] suggest using very powerful machinery such as the seminal pseudorandom generator of Nisan [27]. However, such a PRG does not represent a hash function and implementing it as such would incur a huge computational overhead.

Meanwhile, some papers do indeed consider which concrete hash functions to use. In [15] it was considered to use 2-independent hashing for bottom-$k$ sketches, which was proved in [34] to work for this application. However, bottom-$k$ sketches do not work for SVMs and LSH. Closer to our work, [23] considered the use of 2-independent (and 4-independent) hashing for large-scale classification and online learning with $b$-bit minwise hashing. Their experiments indicate that 2-independent hashing often works, and they state that "the simple and highly efficient 2-independent scheme may be sufficient in practice". However, no amount of experiments can show that this is the case for all input. In fact, we demonstrate in this paper – for the underlying FH and OPH – that this is not the case, and that we cannot trust 2-independent hashing to work in general. As noted, [23] used hashing for similarity estimation in classification, but without considering the quality of the underlying similarity estimation. Due to space restrictions, we do not consider classification in this paper, but instead focus on the quality of the underlying similarity estimation and dimensionality reduction sketches as well as considering these sketches in LSH as the sole applicaton (see also the discussion below).

## 1.2 Our contribution

We analyze the very fast and powerful mixed tabulation scheme of [14] comparing it to some of the most popular and widely employed hash functions. In [14] it was shown that implementing OPH with mixed tabulation gives concentration bounds "essentially as good as truly random". For feature hashing, we first present new concentration bounds for the truly random case improving on [37, 16]. We then argue that mixed tabulation gives essentially as good concentration bounds in the case where the input vectors are not too dense, which is a very common case for applying feature hashing.

Experimentally, we demonstrate that mixed tabulation is almost as fast as the classic multiply-mod-prime hashing scheme. This classic scheme is guaranteed to work well for the considered applications when the data is sufficiently random, but we demonstrate that bias and poor concentration can occur on both synthetic and real-world data. We verify on the same experiments that mixed tabulation has the desired strong concentration, confirming the theory. We also find that mixed tabulation is roughly 40% faster than the very popular MurmurHash3 and CityHash. In our experiments these hash functions perform similar to mixed tabulation in terms of concentration. They do, however, not have the same theoretical guarantees making them harder to trust. We also consider different basic hash functions for implementing LSH with OPH. We demonstrate that the bias and poor concentration of the simpler hash functions for OPH translates into poor concentration for e.g. the recall and number of retrieved data points of the corresponding LSH search structure. Again, we observe that this is not the case for mixed tabulation, which systematically out-performs the faster hash functions. We note that [23] suggests that 2-independent hashing only has problems with dense data sets, but both the real-world and synthetic data considered in this work are sparse or, in the case of synthetic data, can be generalized to arbitrarily sparse data. While we do not consider $b$-bit hashing as in [23], we note that applying the $b$-bit trick to our experiments would only introduce a bias from false positives for all basic hash functions and leave the conclusion the same.

It is important to note that our results do not imply that standard hashing techniques (i.e. multiply-mod prime) never work. Rather, they show that there does exist practical scenarios where the theoretical guarantees matter, making mixed tabulation more consistent. We believe that the very fast evaluation time and consistency of mixed tabulation makes it the best choice for the applications considered in this paper.

## 2 Preliminaries

As mentioned we focus on similarity estimation and feature hashing. Here we briefly describe the methods used. We let $[m] = \{0, \ldots, m-1\}$, for some integer $m$, denote the output range of the hash functions considered.

## 2.1 Similarity estimation

In similarity estimation we are given two sets, $A$ and $B$ belonging to some universe $U$ and are tasked with estimating the Jaccard similarity $J(A, B) = |A \cap B|/|A \cup B|$. As mentioned earlier, this can be solved using $k$ independent repetitions of the MinHash algorithm, however this requires $O(k \cdot |A|)$ running time. In this paper we instead use the faster OPH of Li et al. [22] with the densification scheme of Shrivastava and Li [32]. This scheme works as follows: Let $k$ be a parameter with $k$ being a divisor of $m$, and pick a random hash function $h : U \to [m]$. for each element $x$ split $h(x)$ into two parts $b(x), v(x)$, where $b(x) : U \to [k]$ is given by $h(x) \bmod k$ and $v(x)$ is given by $\lfloor h(x)/k \rfloor$. To create the sketch $S_{OPH}(A)$ of size $k$ we simply let $S_{OPH}(A)[i] = \min_{a \in A, b(a)=i} v(a)$. To estimate the similarity of two sets $A$ and $B$ we simply take the fraction of indices, $i$, where $S_{OPH}(A)[i] = S_{OPH}(B)[i]$.

This is, however, not an unbiased estimator, as there may be *empty bins*. Thus, [31, 32] worked on handling empty bins. They showed that the following addition gives an unbiased estimator with good variance. For each index $i \in [k]$ let $b_i$ be a random bit. Now, for a given sketch $S_{OPH}(A)$, if the $i$th bin is empty we copy the value of the closest non-empty bin going left (circularly) if $b_i = 0$ and going right if $b_i = 1$. We also add $j \cdot C$ to this copied value, where $j$ is the distance to the copied bin and $C$ is some sufficiently large offset parameter. The entire construction is illustrated in Figure 1

| Hash value | 0 1 2 3 | 4 5 6 7 | 8 9 10 11 | 12 13 14 15 | 16 17 18 19 |
|---|---|---|---|---|---|
| Bin | 0 | 1 | 2 | 3 | 4 |
| Value | 0 1 2 3 | 0 1 2 3 | 0 1 2 3 | 0 1 2 3 | 0 1 2 3 |
| h(A) | 0 0 1 1 | 0 1 0 0 | 0 0 0 0 | 1 0 1 0 | 0 0 1 0 |
| S_OPH(A) | 2 | 1 | - | 0 | 2 |

| Bin | 0 | 1 | 2 | 3 | 4 | 5 |
|---|---|---|---|---|---|---|
| Direction | 0 | 1 | 1 | 0 | 0 | 1 |
| S_OPH(A) | 3+C | 2 | 1+2C | 2+2C | 1 | 3 |

Figure 1: Left: Example of one permutation sketch creation of a set $A$ with $|U| = 20$ and $k = 5$. For each of the 20 possible hash value the corresponding bin and value is displayed. The hash values of $A$, $h(A)$, are displayed as an indicator vector with the minimal value per bin marked in red. Note that the 3rd bin is empty. Right: Example of the densification from [32] (right).

## 2.2 Feature hashing

Feature hashing (FH) introduced by Weinberger et al. [37] takes a vector $v$ of dimension $d$ and produces a vector $v'$ of dimension $d' \ll d$ preserving (roughly) the norm of $v$. More precisely, let $h : [d] \rightarrow [d']$ and sgn $: [d] \rightarrow \{-1, +1\}$ be random hash functions, then $v'$ is defined as $v'_i = \sum_{j, h(j)=i} \text{sgn}(j) v_j$. Weinberger et al. [37] (see also [16]) showed exponential tail bounds on $\|v'\|_2^2$ when $\|v\|_\infty$ is sufficiently small and $d'$ is sufficiently large.

## 2.3 Locality-sensitive hashing

The LSH framework of [19] is a solution to the approximate near neighbour search problem: Given a giant collection of sets $\mathcal{C} = A_1, \ldots, A_n$, store a data structure such that, given a query set $A_q$, we can, loosely speaking, efficiently find a $A_i$ with large $J(A_i, A_q)$. Clearly, given the potential massive size of $\mathcal{C}$ it is infeasible to perform a linear scan.

With LSH parameterized by positive integers $K, L$ we create a size $K$ sketch $S_{oph}(A_i)$ (or using another method) for each $A_i \in \mathcal{C}$. We then store the set $A_i$ in a large table indexed by this sketch $T[S_{oph}(A_i)]$. For a given query $A_q$ we then go over all sets stored in $T[S_{oph}(A_q)]$ returning only those that are "sufficiently similar". By picking $K$ large enough we ensure that very distinct sets (almost) never end up in the same bucket, and by repeating the data structure $L$ independent times (creating $L$ such tables) we ensure that similar sets are likely to be retrieved in at least one of the tables.

Recently, much work has gone into providing theoretically optimal [5, 4, 13] LSH. However, as noted in [2], these solutions require very sophisticated locality-sensitive hash functions and are mainly impractical. We therefore choose to focus on more practical variants relying either on OPH [31, 32] or FH [12, 2].

## 2.4 Mixed tabulation

Mixed tabulation was introduced by [14]. For simplicity assume that we are hashing from the universe $[2^w]$ and fix integers $c, d$ such that $c$ is a divisor of $w$. Tabulation-based hashing views each key $x$ as a list of $c$ characters $x_0, \ldots, x_{c-1}$, where $x_i$ consists of the $i$th $w/c$ bits of $x$. We say that the alphabet $\Sigma = [2^{w/c}]$. Mixed tabulation uses $x$ to derive $d$ additional characters from $\Sigma$. To do this we choose $c$ tables $T_{1,i} : \Sigma \rightarrow \Sigma^d$ uniformly at random and let $y = \oplus_{i=0}^{c} T_{1,i}[x_i]$ (here $\oplus$ denotes the XOR operation). The $d$ derived characters are then $y_0, \ldots, y_{d-1}$. To create the final hash value we additionally choose $c + d$ random tables $T_{2,i} : \Sigma \rightarrow [m]$ and define

$$h(x) = \bigoplus_{i \in [c]} T_{2,i}[x_i] \bigoplus_{i \in [d]} T_{2,i+c}[y_i] .$$

Mixed Tabulation is extremely fast in practice due to the word-parallelism of the XOR operation and the small table sizes which fit in fast cache. It was proved in [14] that implementing OPH with mixed tabulation gives Chernoff-style concentration bounds when estimating Jaccard similarity.

Another advantage of mixed tabulation is when generating many hash values for the same key. In this case, we can increase the output size of the tables $T_{2,i}$, and then whp. over the choice of $T_{1,i}$ the resulting output bits will be independent. As an example, assume that we want to map each key to two 32-bit hash values. We then use a mixed tabulation hash function as described above mapping keys to one 64-bit hash value, and then split this hash value into two 32-bit values, which would be

independent of each other with high probability. Doing this with e.g. multiply-mod-prime hashing would not work, as the output bits are not independent. Thereby we significantly speed up the hashing time when generating many hash values for the same keys.

A sample implementation with $c = d = 4$ and 32-bit keys and values can be found below.

```
uint64_t mt_T1[256][4];  // Filled with random bits
uint32_t mt_T2[256][4];  // Filled with random bits

uint32_t mixedtab(uint32_t x) {
  uint64_t h=0;  // This will be the final hash value
  for(int i = 0;i < 4;++i, x >>= 8)
    h ^= mt_T1[(uint8_t)x][i];
  uint32_t drv=h >> 32;
  for(int i = 0;i < 4;++i, drv >>= 8)
    h ^= mt_T2[(uint8_t)drv][i];
  return (uint32_t)h;
}
```

The main drawback to mixed tabulation hashing is that it needs a relatively large random seed to fill out the tables $T_1$ and $T_2$. However, as noted in [14] for all the applications we consider here it suffices to fill in the tables using a $\Theta(\log |U|)$-independent hash function.

## 3   Feature Hashing with Mixed Tabulation

As noted, Weinberger et al. [37] showed exponential tail bounds for feature hashing. Here, we first prove improved concentration bounds, and then, using techniques from [14] we argue that these bounds still hold (up to a small additive factor polynomial in the universe size) when implementing FH with mixed tabulation.

The concentration bounds we show are as follows (proved in the full version).

**Theorem 1.** *Let $v \in \mathbb{R}^d$ with $\|v\|_2 = 1$ and let $v'$ be the $d'$-dimensional vector obtained by applying feature hashing implemented with truly random hash functions. Let $\varepsilon, \delta \in (0, 1)$. Assume that $d' \geq 16\varepsilon^{-2} \lg(1/\delta)$ and $\|v\|_\infty \leq \frac{\sqrt{\varepsilon \log(1+\frac{4}{\varepsilon})}}{6\sqrt{\log(1/\delta)\log(d'/\delta)}}$. Then it holds that*

$$\Pr\left[1 - \varepsilon < \|v'\|_2^2 < 1 + \varepsilon\right] \geq 1 - 4\delta. \tag{1}$$

Theorem 1 is very similar to the bounds on feature hashing by Weinberger et al. [37] and Dasgupta et al. [16], but improves on the requirement on the size of $\|v\|_\infty$. Weinberger et al. [37] show that (1) holds if $\|v\|_\infty$ is bounded by $\frac{\varepsilon}{18\sqrt{\log(1/\delta)\log(d'/\delta)}}$, and Dasgupta et al. [16] show that (1) holds if $\|v\|_\infty$ is bounded by $\sqrt{\frac{\varepsilon}{16\log(1/\delta)\log^2(d'/\delta)}}$. We improve on these results factors of $\Theta\left(\sqrt{\frac{1}{\varepsilon}\log(1/\varepsilon)}\right)$ and $\Theta\left(\sqrt{\log(1/\varepsilon)\log(d'/\delta)}\right)$ respectively. We note that if we use feature hashing with a pre-conditioner (as in e.g. [16, Theorem 1]) these improvements translate into an improved running time.

Using [14, Theorem 1] we get the following corollary.

**Corollary 1.** *Let $v, \varepsilon, \delta$ and $d'$ be as in Theorem 1, and let $v'$ be the $d'$-dimensional vector obtained using feature hashing on $v$ implemented with mixed tabulation hashing. Then, if $\mathrm{supp}(v) \leq |\Sigma|/(1 + \Omega(1))$ it holds that*

$$\Pr\left[1 - \varepsilon < \|v'\|_2^2 < 1 + \varepsilon\right] \geq 1 - 4\delta - O\left(|\Sigma|^{1 - \lfloor d/2 \rfloor}\right).$$

In fact Corollary 1 holds even if both $h$ and $\mathrm{sgn}$ from Section 2.2 are implemented using the same hash function. I.e., if $h^\star : [d] \to \{-1, +1\} \times [d']$ is a mixed tabulation hash function as described in Section 2.4.

We note that feature hashing is often applied on very high dimensional, but sparse, data (e.g. in [2]), and thus the requirement $\mathrm{supp}(v) \leq |\Sigma|/(1 + \Omega(1))$ is not very prohibitive. Furthermore, the target

dimension $d'$ is usually logarithmic in the universe, and then Corollary 1 still works for vectors with polynomial support giving an exponential decrease.

## 4   Experimental evaluation

We experimentally evaluate several different basic hash functions. We first perform an evaluation of running time. We then evaluate the fastest hash functions on synthetic data confirming the theoretical results of Section 3 and [14]. Finally, we demonstrate that even on real-world data, the provable guarantees of mixed tabulation sometimes yields systematically better results.

Due to space restrictions, we only present some of our experiments here, and refer to the full version for more details.

We consider some of the most popular and fast hash functions employed in practice in $k$-wise PolyHash [10], Multiply-shift [17], MurmurHash3 [6], CityHash [29], and the cryptographic hash function Blake2 [7]. Of these hash functions only mixed tabulation (and very high degree PolyHash) *provably* works well for the applications we consider. However, Blake2 is a cryptographic function which provides similar guarantees conditioned on certain cryptographic assumptions being true. The remaining hash functions have provable weaknesses, but often work well (and are widely employed) in practice. See e.g. [1] who showed how to break both MurmurHash3 and Cityhash64.

All experiments are implemented in C++11 using a random seed from http://www.random.org. The seed for mixed tabulation was filled out using a random 20-wise PolyHash function. All keys and hash outputs were 32-bit integers to ensure efficient implementation of multiply-shift and PolyHash using Mersenne prime $p = 2^{61} - 1$ and GCC's 128-bit integers.

We perform two time experiments, the results of which are presented in Table 1. Namely, we evaluate each hash function on the same $10^7$ randomly chosen integers and use each hash function to implement FH on the News20 dataset (discussed later). We see that the only two functions faster than mixed tabulation are the very simple multiply-shift and 2-wise PolyHash. MurmurHash3 and CityHash were roughly 30-70% slower than mixed tabulation. This even though we used the official implementations of MurmurHash3, CityHash and Blake2 which are highly optimized to the x86 and x64 architectures, whereas mixed tabulation is just standard, portable C++11 code. The cryptographic hash function, Blake2, is orders of magnitude slower as we would expect.

Table 1: Time taken to evaluate different hash functions to 1) hash $10^7$ random numbers, and 2) perform feature hashing with $d' = 128$ on the entire News20 data set.

| Hash function | time $(1..10^7)$ | time (News20) |
|---|---|---|
| Multiply-shift | 7.72 ms | 55.78 ms |
| 2-wise PolyHash | 17.55 ms | 82.47 ms |
| 3-wise PolyHash | 42.42 ms | 120.19 ms |
| MurmurHash3 | 59.70 ms | 159.44 ms |
| CityHash | 59.06 ms | 162.04 ms |
| Blake2 | 3476.31 ms | 6408.40 ms |
| Mixed tabulation | 42.98 ms | 90.55 ms |

Based on Table 1 we choose to compare mixed tabulation to multiply-shift, 2-wise PolyHash and MurmurHash3. We also include results for 20-wise PolyHash as a (cheating) way to "simulate" truly random hashing.

### 4.1   Synthetic data

For a parameter, $n$, we generate two sets $A, B$ as follows. The intersection $A \cap B$ is created by sampling each integer from $[2n]$ independently at random with probability $1/2$. The symmetric difference is generated by sampling $n$ numbers greater than $2n$ (distributed evenly to $A$ and $B$). Intuitively, with a hash function like $(ax + b) \bmod p$, the dense subset of $[2n]$ will be mapped very systematically and is likely (i.e. depending on the choice of $a$) to be spread out evenly. When using

OPH, this means that elements from the intersection is more likely to be the smallest element in each bucket, leading to an over-estimation of $J(A, B)$.

We use OPH with densification as in [32] implemented with different basic hash functions to estimate $J(A, B)$. We generate one instance of $A$ and $B$ and perform 2000 independent repetitions for each different hash function on these $A$ and $B$. Figure 2 shows the histogram and mean squared error (MSE) of estimates obtained with $n = 2000$ and $k = 200$. The figure confirms the theory: Both multiply-shift and 2-wise PolyHash exhibit bias and bad concentration whereas both mixed tabulation and MurmurHash3 behaves essentially as truly random hashing. We also performed experiments with $k = 100$ and $k = 500$ and considered the case of $n = k/2$, where we expect many empty bins and the densification of [32] kicks in. All experiments obtained similar results as Figure 2.

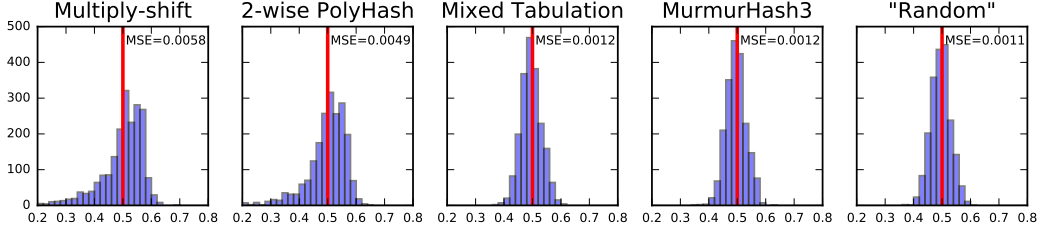

Figure 2: Histograms of set similarity estimates obtained using OPH with densification of [32] on synthetic data implemented with different basic hash families and $k = 200$. The mean squared error for each hash function is displayed in the top right corner.

For FH we obtained a vector $v$ by taking the indicator vector of a set $A$ generated as above and normalizing the length. For each hash function we perform 2000 independent repetitions of the following experiment: Generate $v'$ using FH and calculate $\|v'\|_2^2$. Using a good hash function we should get good concentration of this value around 1. Figure 3 displays the histograms and MSE we obtained for $d' = 200$. Again we see that multiply-shift and 2-wise PolyHash give poorly concentrated results, and while the results are not biased this is only because of a very heavy tail of large values. We also ran experiments with $d' = 100$ and $d' = 500$ which were similar.

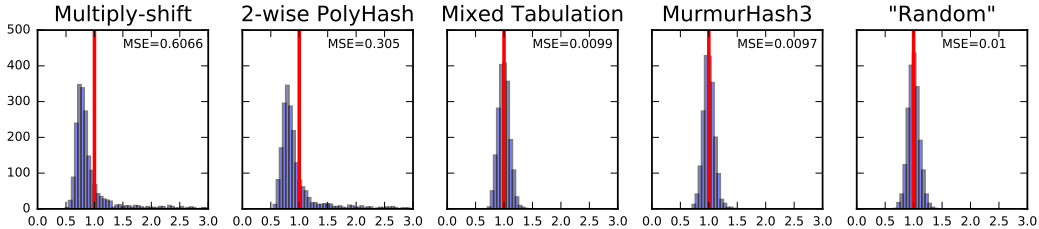

Figure 3: Histograms of the 2-norm of the vectors output by FH on synthetic data implemented with different basic hash families and $d' = 200$. The mean squared error for each hash function is displayed in the top right corner.

We briefly argue that this input is in fact quite natural: When encoding a document as shingles or bag-of-words, it is quite common to let frequent words/shingles have the lowest identifier (using fewest bits). In this case the intersection of two sets $A$ and $B$ will likely be a dense subset of small identifiers. This is also the case when using Huffman Encoding [18], or if identifiers are generated on-the-fly as words occur. Furthermore, for images it is often true that a pixel is more likely to have a non-zero value if its neighbouring pixels have non-zero values giving many consecutive non-zeros.

**Additional synthetic results** We also considered the following synthetic dataset, which actually showed even more biased and poorly concentrated results. For similarity estimation we used elements from $[4n]$, and let the symmetric difference be uniformly random sampled elements from $\{0 \ldots, n - 1\} \cup \{3n, \ldots, 4n - 1\}$ with probability $1/2$ and the intersection be the same but for $\{n, \ldots, 3n - 1\}$. This gave an MSE that was rougly 6 times larger for multiply-shift and 4 times larger for 2-wise

PolyHash compared to the other three. For feature hashing we sampled the numbers from $0$ to $3n - 1$ independently at random with probability $1/2$ giving an MSE that was 20 times higher for multiply-shift and 10 times higher for 2-wise PolyHash.

We also considered both datasets without the sampling, which showed an even wider gap between the hash functions.

## 4.2 Real-world data

We consider the following real-world data sets

- **MNIST** [21] Standard collection of handwritten digits. The average number of non-zeros is roughly 150 and the total number of features is 728. We use the standard partition of 60000 database points and 10000 query points.
- **News20** [11] Collection of newsgroup documents. The average number of non-zeros is roughly 500 and the total number of features is roughly $1.3 \cdot 10^6$. We randomly split the set into two sets of roughly 10000 database and query points.

These two data sets cover both the sparse and dense regime, as well as the cases where each data point is similar to many other points or few other points. For MNIST this number is roughly 3437 on average and for News20 it is roughly 0.2 on average for similarity threshold above $1/2$.

**Feature hashing** We perform the same experiment as for synthetic data by calculating $\|v'\|_2^2$ for each $v$ in the data set with 100 independent repetitions of each hash function (i.e. getting $6,000,000$ estimates for MNIST). Our results are shown in Figure 4 for output dimension $d' = 128$. Results with $d' = 64$ and $d' = 256$ were similar. The results confirm the theory and show that mixed tabulation

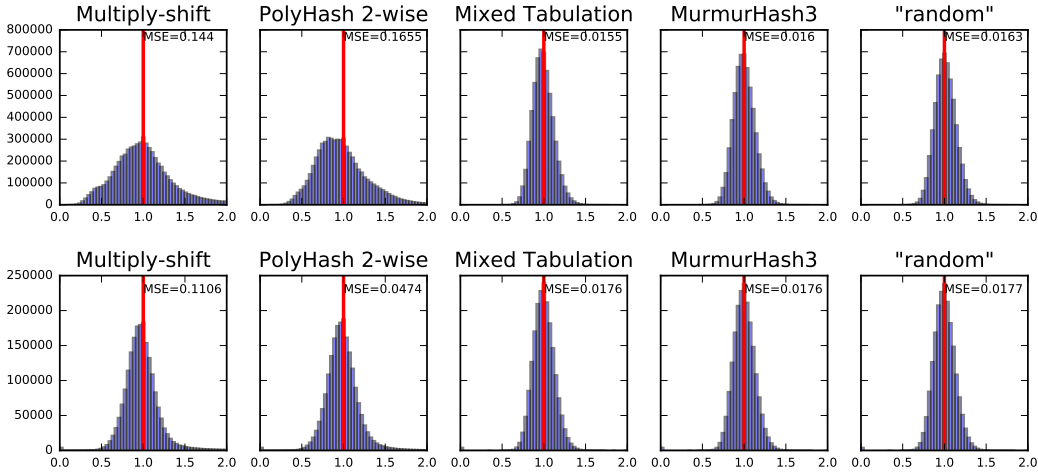

Figure 4: Histograms of the norm of vectors output by FH on the MNIST (top) and News20 (bottom) data sets implemented with different basic hash families and $d' = 128$. The mean squared error for each hash function is displayed in the top right corner.

performs essentially as well as a truly random hash function clearly outperforming the weaker hash functions, which produce poorly concentrated results. This is particularly clear for the MNIST data set, but also for the News20 dataset, where e.g. 2-wise Polyhash resulted in $\|v'\|_2^2$ as large as 16.671 compared to 2.077 with mixed tabulation.

**Similarity search with LSH** We perform a rigorous evaluation based on the setup of [31]. We test all combinations of $K \in \{8, 10, 12\}$ and $L \in \{8, 10, 12\}$. For readability we only provide results for multiply-shift and mixed tabulation and note that the results obtained for 2-wise PolyHash and MurmurHash3 are essentially identical to those for multiply-shift and mixed tabulation respectively.

Following [31] we evaluate the results based on two metrics: 1) The fraction of total data points retrieved per query, and 2) the *recall* at a given threshold $T_0$ defined as the ratio of retrieved data

points having similarity at least $T_0$ with the query to the total number of data points having similarity at least $T_0$ with the query. Since the recall may be inflated by poor hash functions that just retrieve many data points, we instead report #retrieved/recall-ratio, i.e. the number of data points that were retrieved divided by the percentage of recalled data points. The goal is to minimize this ratio as we want to simultaneously retrieve few points and obtain high recall. Due to space restrictions we only report our results for $K = L = 10$. We note that the other results were similar.

Our results can be seen in Figure 5. The results somewhat echo what we found on synthetic data. Namely, 1) Using multiply-shift overestimates the similarities of sets thus retrieving more points, and 2) Multiply-shift gives very poorly concentrated results. As a consequence of 1) Multiply-shift does, however, achieve slightly higher recall (not visible in the figure), but despite recalling slightly more points, the #retrieved / recall-ratio of multiply-shift is systematically worse.

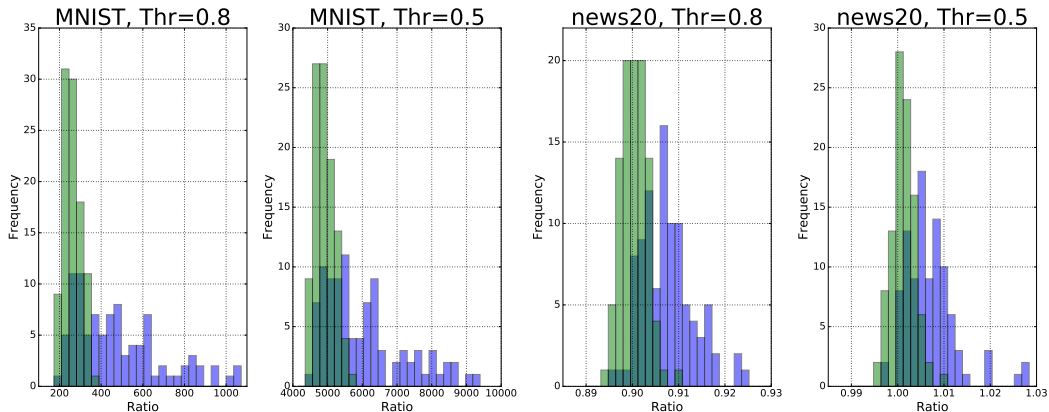

Figure 5: Experimental evaluation of LSH with OPH and different hash functions with $K = L = 10$. The hash functions used are multiply-shift (blue) and mixed tabulation (green). The value studied is the retrieved / recall-ratio (lower is better).

## 5    Conclusion

In this paper we consider mixed tabulation for computational primitives in computer vision, information retrieval, and machine learning. Namely, similarity estimation and feature hashing. It was previously shown [14] that mixed tabulation provably works essentially as well as truly random for similarity estimation with one permutation hashing. We complement this with a similar result for FH when the input vectors are sparse, even improving on the concentration bounds for truly random hashing found by [37, 16].

Our empirical results demonstrate this in practice. Mixed tabulation significantly outperforms the simple hashing schemes and is not much slower. Meanwhile, mixed tabulation is 40% faster than both MurmurHash3 and CityHash, which showed similar performance as mixed tabulation. However, these two hash functions do not have the same theoretical guarantees as mixed tabulation. We believe that our findings make mixed tabulation the best candidate for implementing these applications in practice.

## Acknowledgements

The authors gratefully acknowledge support from Mikkel Thorup's Advanced Grant *DFF-0602-02499B* from the *Danish Council for Independent Research* as well as the *DABAI project*. Mathias Bæk Tejs Knudsen gratefully acknowledges support from the *FNU project AlgoDisc*.

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
