[Reviews · NeurIPS 2017]

Reviewer 1



The paper looks into a very basic and important area of the choice of hash functions for feature hashing and densified one permutation hashing. The paper empirically shows that mixed tabulation is superior, in terms of speed, over popularly used murmurhash3 and both are better in accuracy that shift-multiply. The main contribution of the paper is an important empirical comparison of famous hashing schemes of similarity estimation and retrieval of feature hashing and densified hashing, which will be a good resource for practitioners. Even though the contributions are mostly empirical since murmurhash3 is widely popular this paper will have some impact in practice.

Reviewer 2



The main contribution of the paper is the set of empirical comparisons of various hashing functions, mutiply, 2-universal (2U), Murmur, and mixed tabulation, for (1) similarity estimation and (2) LSH. This type of empical evaluations is very important and will benefit practioners. Overall, this is a good paper. For improving the paper, I would suggest the authors to take into consideration of the following comments (some of which might be crucial): ---1. There is already a study of using 2U and 4U hashing for similarity estimation and classification tasks. See [1] b-Bit Minwise Hashing in Practice. Proceedings of the 5th Asia-Pacific Symposium on Internetware, 2013. [2] https://arxiv.org/abs/1205.2958, which is a more detailed version of [1], as far as I can tell. [1,2] used 2U and 4U for evalating both b-bit minwise hashing and FH (although they used VW to refer to FH). For example, [1,2] showed that for dense sets, 2U and 4U had an obvious bias and larger MSE. I would suggest the authors to cite one of [1,2] and comment on the additional contributions beyond [1,2]. ---2. [1,2] already showed that for classification tasks, 2U and 4U seems to be sufficiently accurate compared to using fully random hash functions. This paper ommited classification experiments. It is not at all surprising that mixed tabulation will work as well as fully random hash for classification, which the NIPS community would probably care more (than LSH). ---3. This paper used 2U instead of 4U. It would be more coninvincing to present the results using 4U (or both 2U and 4U) as it is well-understood that in some cases 4U is indeed better. ---4. The LSH experiments can have been presented more convincingly. a) In practice, we typically must guaranttee a good recall, even at the cost of retrieving more points. therefore, while the presentation of using the ratio # retrieved / recall is useful, it may cover important details. b) only mulply hash and mix tabulation results are presented. It is important to also present fully random results and 2U/4U hashing results. c) If the results from different hash functions as shown in Figure 5 really differ that much, then it is not that appropriate to compare the results at a fixed K and L, because each hash function may work better with a particular parameter. d) The LSH parameters are probably chosen not so appropriately. The paper says it followed [30] for the parameters K,L by using K ={8,10,12} and L={8,10,12}. However, [30] actually used for K = {6, 8, 10, 12} and L = {4, 8, 16, 32, 64, 128} e) LSH-OPH with K = {8,10,12} is likely too large. L =[8,10,12} is likely too small, according to prior experience. If the parameters are not (close to) optimal, then it is difficult to judge usefulness of the experiment results. ----------- In Summary, overall, the topic of this paper is very important. The limitation of the current submission includes i) a lack of citation of the prior work [1,2] and explanation on the additional contribution beyond [1,2]. ii) the classification experiments are missing. It is very possible that no essential difference will be observed for any hash function, as concluded in the prior work [1,2]. iii) the LSH experiments may have issues and the conclusions drawn from the LSH experiments are hard to judge.

Reviewer 3



This paper considers the mixed tabulation hash function of [DKRT] in FOCS 2015 which had been proved to perform like a true random function in some applications (like Jaccard Similarity using One Permutation Hashing). This paper provides similar theoretical results for Feature Hashing and more importantly experiments to support this theoretical results. They use mixed tabulation for the OPH and Feature Hashing and compared its performance to a variety of hash functions. They showed that for the hash functions whose results were comparable, mixed tabulation was at least 40% faster. I believe this is a nice experimental result supported by a solid theory. However, the paper is not very well written and it has many typos. There are also certain places that are not clear, for example Figure 1 is not well explained and it is not quite clear what is going on from the text. minor comments: abstract, functions -> function page 2, fast the -> fast as the page 2, they -> They page 3, with k a divisor of -> with k being a divisor of page 3, simple -> simply page 5, some the most -> some of the most page 11, theorem 2 should be changed to theorem 1